# Hormones and Heterosis in Hybrid Balsam Poplar (*Populus balsamifera* L.)

**Yue Hu *** and **Barb R. Thomas**

Department of Renewable Resources, 442 Earth Science Building, University of Alberta, Edmonton, AB T6G 2E3, Canada; bthomas@ualberta.ca
* Correspondence: yhu6@ualberta.ca; Tel.: +1-403-992-2640

**Abstract:** Balsam poplar (*Populus balsamifera* L.) is a transcontinental tree species in North America, making it an ideal species to study intra-specific hybrid vigour as a tool for increasing genetic gain in growth. We tested the hypothesis that intra-specific breeding of disparate populations of balsam poplar would lead to the expression of hybrid vigour and we determined the role of endogenous hormones linked to ecophysiological and growth performance. In September 2009, three field trials were established in Canada (two in Alberta (AB), i.e., Fields AB1 and AB2, and one in Quebec (QC), i.e., Field QC1) in conjunction with Alberta-Pacific Forest Industries Inc. and the Ministry of Forests, Wildlife and Parks, Quebec. Five male parents from each province as well as five female parents from QC and four female parents from AB were used for breeding intra-regional and inter-regional crosses. Based on a significant difference at year six for height and diameter, from the AB1 and AB2 field trials, the AB × QC cross-type was selected for further study. Cuttings from the AB × QC cross-type were grown in a randomized complete block design under near-optimal greenhouse conditions. Families were identified as slow- or fast-growing, and the relationship between hormone levels and growth performance of the genotypes within the families were examined. In late June, after 34 days of growth, internode tissue samples collected from each progeny were analyzed for gibberellic acids, indole-3-acetic acid, and abscisic acid content. Stem volume of two-month-old rooted cuttings, grown under optimal greenhouse conditions, was positively and significantly correlated with the photosynthetic rate, greenhouse growth, and stem volume of 8-year-old field-grown trees (Fields AB1 values: r = 0.629 and *p* = 0.012; AB2 values: r = 0.619 and *p* = 0.014, and QC1 values: r = 0.588 and *p* = 0.021, respectively). We determined that disparate and native populations of balsam poplar can be bred to produce superior progeny with enhanced stem growth traits.

**Keywords:** balsam poplar; disparate populations; hybrid vigour; plant physiology

## 1. Introduction

Heterosis, or hybrid vigour, refers to the phenomenon in which hybrids outperform their parents in yield, biomass, biotic and abiotic stress tolerance, or other traits [1]. Hybrid vigour, typically achieved through the controlled crossing of two species, or pure genetic lines of the same species, has long been exploited in agriculture [2–4] and in some tree species including *Populus* [5]. For poplars, including the aspens, two or more species are typically crossed to produce hybrid progeny, some of which can be expected to yield growth performance far superior than either parent (i.e., hybrid vigour or heterosis). Balsam poplar (*Populus balsamifera* L.) is a transcontinental species in North America ranging from Alaska to Newfoundland [6]. This species occupies a wide range of climatic and site conditions and often grows in mixed stands with conifers or other broadleaf trees, contributing to stand and landscape level diversity [6]. This wide range makes it an ideal species to study within-species hybrid vigour as a tool for increasing genetic gain in volume for a given population and rotation

age [7,8]. Because of the clonal nature of this species and its ease of propagation from cuttings, any volume gain achieved through hybrid vigour can be rapidly exploited in a tree improvement program through cloning the superior individuals [9].

Knowledge regarding the physiological basis of heterosis is sporadic and has mainly focused on specific traits, such as freezing tolerance in *Arabidopsis* [10]. In addition, however, there have also been studies on the role of gibberellic acids (GAs) in the regulation of heterosis (e.g., [11]). The majority of GA metabolism genes in various plant species have been identified and characterized [12,13]. Gibberellins, a group of tetracyclic diterpenoid compounds, function as plant hormones that play an important role in the regulation of many aspects of plant growth and development, such as seed germination, stem elongation, leaf expansion, flower and fruit development, and wood formation [14]. Gibberellins residing in three different cellular compartments (plastid, endoplasmic reticulum, and cytoplasm), being synthesized via the terpenoid pathway, require terpene synthase (TPSs), cytochrome P450 mono-oxygenase (P450s), and 2-oxoglutarate dependent dehydrogenase (2 ODDs) for the biosynthesis of bioactive GA from geranylgeranyl diphosphate (GGDP) in plants [15]. The GA 20-oxidases (*GA20ox*) and GA 3-oxidases (*GA3ox*) convert the early GA structures ($GA_{12}$ or $GA_{53}$) to the growth-active structures, $GA_4$ and $GA_1$, via the early non-13 hydroxylation pathway (leading to $GA_4$) and the early 13-hydroxylation pathway (leading to $GA_1$), respectively [11]. The inactivation of these growth-activating GAs and many of their early precursors is achieved by GA 2-oxidases (*GA2ox*) [16].

Recent work in rice has revealed evidence that endogenous gibberellins play a key role regulating heterosis [11]. Moreover, Rood et al. [17] analyzed the differences in responsiveness to the exogenous application of $GA_3$ and endogenous levels of GAs between F1 hybrids and their inbred parents of diallele combinations in maize (*Zea mays*). They concluded that the increased endogenous concentration of GA in the hybrids could provide a phytohormone basis for heterosis for shoot growth. Additionally, Park et al. [14] compared the growth of young conifer seedlings under optimal conditions with the field performance of the same seedlings via a retrospective approach and found that endogenous GA levels may explain much of the natural variation seen in tree stem size in even-aged pine forests.

Indole-3-acetic acid (IAA), the most common plant hormone of the auxin class, has been linked to tree stem radial growth both for conifers [18] and deciduous trees [19]. IAA regulates various aspects of plant growth and development and acts as a positional signal that controls cambial growth rate by regulating the radial number of dividing cells in the cambial meristem, which is an important component in determining cambial growth rate [20]. Overall, IAA promotes cell division, elongation, and differentiation, whereas abscisic acid (ABA) regulates IAA biosynthesis and activity [21]. Abscisic acid, a sesquiterpene, plays an important role in seed development and maturation and induces dormancy in buds, underground stem, and seeds [22]. ABA is also called a stress hormone because the production of hormones is stimulated by drought, water logging, and other adverse environmental conditions by regulating stomatal closure, inducing the expression of stress responsive genes and the accumulation of osmo-compatible solutes [23].

In order to test the role of endogenous hormones including links to physiological and growth performance in balsam poplar produced from breeding disparate populations, a greenhouse study was designed to grow progeny from selected families under near-optimal growing conditions. The stem tissue was harvested, prior to any senescence or corresponding hormone degradation, for GA, IAA, and ABA analysis in order to examine whether there is a causal relationship between hormone concentration in the elongating stem (internodes) tissue and growth rate of the vegetatively propagated progeny from selected families of balsam poplar.

## 2. Materials and Methods

### 2.1. Selection of Families and Progeny

In order to test the hypothesis that within-species breeding will lead to the expression of hybrid vigour, a series of controlled crosses were completed including local × local and local × distant parental types from both Alberta (AB) and Quebec (QC) sources of balsam poplar. Five male parents from each province (Abitibi, QC and Athabasca, AB regions) as well as five female parents from QC and four female parents from AB were used for breeding, both for within-region and between-region crosses. Parent trees were identified in AB and QC and bred in the winter of 2005, the seedlings were grown in stool-beds, and cuttings were taken to establish three field trials in September 2009. In each field trial trees were planted in 4-tree family plots, with 10 blocks at a 2.5 × 2.5 m spacing on two sites (Fields AB1 and AB2) in AB (Alberta-Pacific Forest Industries Inc. (Al-Pac) millsite, 54° N, 112° W, 575 m, mean annual precipitation of 458 mm [24]), and a single site (Field QC1) in QC located at Trécesson, (48° N, 78° W, 348 m, mean annual precipitation of 890 mm [25]). The crosses produced a total of 33 families of AB × AB, AB × QC, QC × AB, and QC × QC cross-types, respectively. Each progeny from each family is represented once in each trial across the 10 blocks. In summer 2016, a greenhouse trial using families identified as slow- and fast-growing in cross-type AB × QC were selected in order to use extremes of performance for examining the relationship between hormone levels and growth performance of the selected genotypes within specific families.

Stem volumes were calculated using the 6-year-old tree data from AB1 and AB2 field sites. The relationship between individual progeny and stem volume for the AB × QC cross-types was then plotted (Figure 1). Across the eight families in this cross-type, there were distinctly different patterns of stem volume in the progeny. Three groups with three progeny per family were selected as follows: (1) a fast-growing (FG) group (families AP5396, AP5402, and AP 5416); (2) a slow-growing (SG) group (families AP5401, AP5411, and AP5414); and (3) three slow-growing progeny selected from the fast-growing families (SFG) (families AP5396, AP5402, and AP 5416). The selection of these progeny and families allowed for the following comparisons: slow-growing vs. fast-growing within the same family group, and slow-growing vs. slow-growing between different family groups. In total, 27 individual progeny were selected for study within each family and group type within the AB × QC cross-type (Table 1).

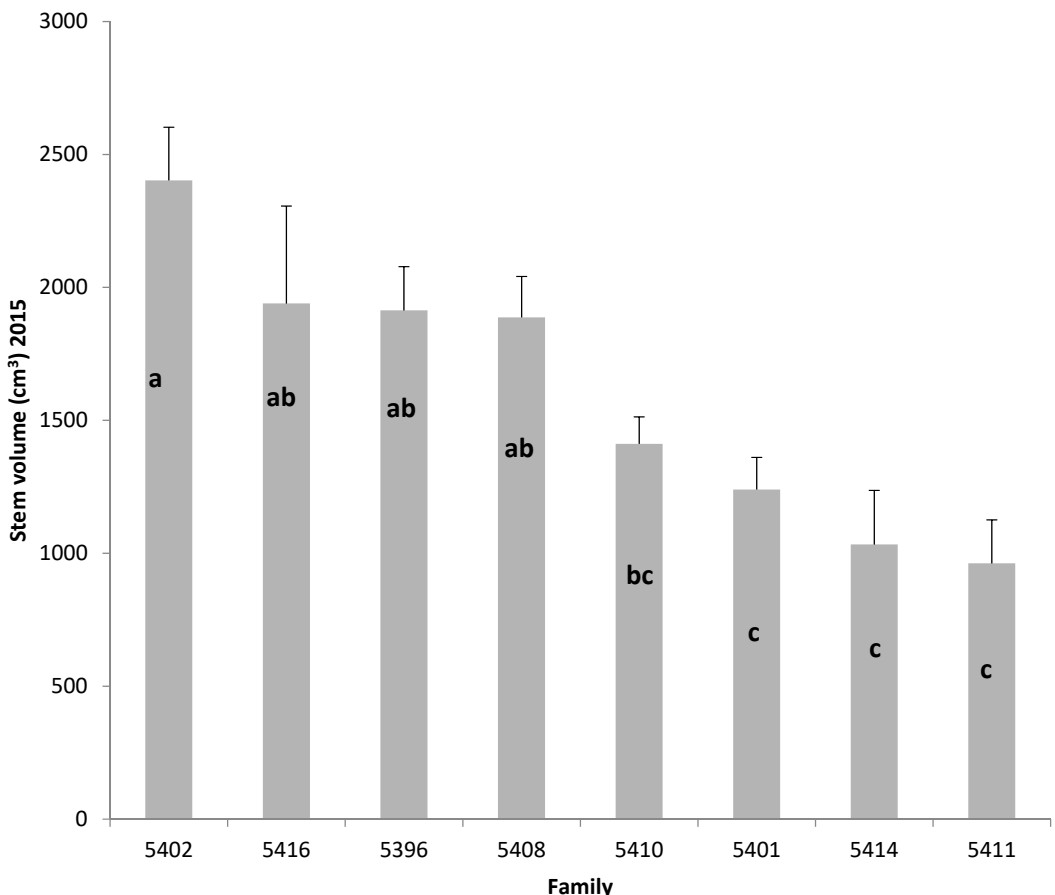

**Figure 1.** Mean stem volumes (cm³) (+SE) for 6-year-old Alberta × Quebec cross-type families grown at two Alberta sites (AB1 and AB2). Significant differences between family means are indicated by different letters at $p \leq 0.05$.

**Table 1.** Family codes, and selected progeny for each group type, fast-growing (FG), slow-growing (SG), and slow fast-growing (SFG), based on 6-year-old growth from two field sites (AB1 and AB2).

| Family | Fast-Growing Progeny | Group (Fast-Growing (FG)) | Slow-Growing Progeny | Group (Slow-Growing in Fast-Growing Group (SFG) or Slow-Growing (SG) Group) |
|---|---|---|---|---|
| AP5396 | 147071 | FG | 147043 | SFG |
|  | 147083 | FG | 147051 | SFG |
|  | 147072 | FG | 147041 | SFG |
| AP5401 |  |  | 178092 | SG |
|  |  |  | 178073 | SG |
|  |  |  | 178104 | SG |
| AP5402 | 180071 | FG | 180093 | SFG |
|  | 180103 | FG | 180063 | SFG |
|  | 180081 | FG | 180094 | SFG |
| AP5411 |  |  | 255081 | SG |
|  |  |  | 255011 | SG |
|  |  |  | 255102 | SG |
| AP5414 |  |  | 270051 | SG |
|  |  |  | 270101 | SG |
|  |  |  | 270061 | SG |
| AP5416 | 272084 | FG | 272024 | SFG |
|  | 272102 | FG | 272071 | SFG |
|  | 272091 | FG | 272023 | SFG |

## 2.2. Greenhouse Propagation

Dormant branch cuttings that were 40–50 cm long were collected on March 31, 2016 from the selected 27 progeny located in the AB1 field at the Al-Pac mill site, placed in black plastic bags and stored in the freezer for eight days at −4 °C, until the greenhouse experiment started. Cuttings were soaked in cold water for two days at room temperature in the lab without the use of any additional rooting hormone [26], with water replaced daily with fresh cold water. On April 11, eight stem cuttings 6–9 cm long with a minimum of two buds for each of the 27 progeny were rooted in Format 360 Hillsons Rootrainers trays (Beaver Plastics Ltd., Acheson, AB, Canada) filled with Sunshine Mix #3 (Sun Gro Horticulture, Vancouver, BC, Canada). Budburst was scored and recorded three times over the course of one week. Upon bud flush, and after height growth of approximately 10 cm was reached (~Day 40 after striking), cuttings were transplanted directly into 2 L pots filled with Sunshine Mix #4 (Sun Gro Horticulture, Vancouver, BC, Canada). The rooted cuttings (stecklings) were grown in the greenhouse at the University of Alberta and Day 1 of the experiment started on 24 May 2016 under natural light supplemented by cool-white fluorescent lamps to provide a 21 h long photoperiod and a minimum photosynthetic photon flux density (PPFD) of 400 µmol m$^{-2}$ s$^{-1}$ at plant level. Maximum day and night temperatures were maintained at approximately 25 °C and 18 °C, respectively, throughout the experimental period. The stecklings were kept well-watered and fertilized using a 20-20-20 commercial water-soluble fertilizer (20:20:20 plus micronutrients (Fe 0.1%; Mn 0.05%; Zn 0.05%; Cu 0.05%; B 0.02%; Mo 0.0005%)) (Plant Products Co. Ltd. Brampton, ON, Canada) at a pH of 5.8–6.3, adjusted by adding phosphoric acid (H$_3$PO$_4$). The greenhouse was well ventilated and PPFD, humidity, and air temperature were continuously monitored and recorded using a HOBO U12-012 data logger (Onset Computer Cooperation, Pocasset, MA, USA). The potted stecklings were rotated weekly to minimize position effect in the greenhouse [27].

## 2.3. Measurements, Harvest and Selection for Hormone Analysis

Caliper, measured at the base of the new stem, and height growth were measured every 10 days after transplanting, starting on Day 1 of the experiment (24 May 2016). Gas exchange measurements including photosynthetic rate (A), stomatal conductance (Gs), and intrinsic water use efficiency (iWUE) were made using a CIRAS-3 infrared gas analyzer (IRGA) (PP Systems, Amesbury, MA, USA) and a broad leaf cuvette (PLC4 (B) Broad Leaf Cuvette, PP Systems, Amesbury, MA, USA), the cuvette window being 18 mm diameter in size with a total area of 2.5 cm$^2$, just prior to harvesting on Day 34, 27 June 2016 (near the longest day of the year). In all, 15 progeny were selected (six from FG, six from SFG, and three from SG) from three families for further hormone analysis.

At harvest, 8 ramets of each of the 15 progeny (genotypes) were grouped based on a visual assessment of vigour, which in turn was based on height and diameter, to select: (a) the largest, (b) the second largest, and (c) the third largest ramet for use in the hormone level analysis in the elongating internode stem tissue [14].

For each of the selected ramets, leaves were snipped off at the base of their petiole, along the chosen length of stem while the tree was still intact. The upper 20–30% (all internodes which were still elongating plus two lower internodes which had ceased to elongate) of the stem were harvested, wrapped in a double-layer of aluminum foil forming a package, and placed onto dry ice for storage prior to preparation for the hormone analysis. The roots were then carefully washed and put into paper bags. The biomass components (leaves, remaining stem tissue, and roots) were stored in paper bags before drying for two days at 65 °C, then measured using a model AV53 scale (readability 0.001 g, OHAUS Adventurer Pro, Melrose, MA, USA). After drying, the leaves were ground in a ball grinder (Model MM200, Retsch Inc., Haan, North Rhine-Westphalia, Germany) and stored in 20 mL plastic scintillation vials (Fisher Scientific, Hampton, NH, USA) in preparation for δ$^{13}$C analysis at the University of Alberta's Natural Resources Analytical Laboratory (NRAL). After five days in the −80 °C freezer, the stem samples, which were harvested earlier for hormone analysis, were freeze-dried in



a FreeZone®2.5 L Benchtop freeze dry system (Labconco Corporation, Kansas City, MO, USA) for three days.

## 2.4. Analysis of GAs, IAA, and ABA

One gram dry weight (DW) of each tissue sample was ground with liquid $N_2$ and washed sea sand (Fisher Scientific, Fair Lawn, NJ, USA), then extracted in 80% MeOH ($H_2O$:MeOH = 20:80, *v/v*). Following this, 250 ng [$^{13}C_6$] IAA (gift from Dr. J. Cohen, available from Cambridge Isotope Laboratories, Inc., Tewksbury, MA, USA), 200 ng [$^2H_6$] ABA (a gift from Drs. L. Rivier and M. Saugy, University of Lausanne, Lausanne, Switzerland), and 20–40 ng each of [$^2H_2$] $GA_{15}$, $GA_{24}$, $GA_9$, $GA_{20}$, $GA_4$, $GA_1$, $GA_8$, and $GA_{34}$ (deuterated GAs were obtained from Professor L.N. Mander, Research School of Chemistry, Australian National University, Canberra, Australia) were added to the aqueous MeOH extraction solvent as internal standards. Subsequent purification, separation, and stable isotope dilution analysis by gas chromatography–mass spectrometry (GC-MS) selected ion monitoring (SIM) were accomplished as described by Kurepin et al. [28].

## 2.5. Data Analysis

All the growth data (height, caliper, and stem volume) were analyzed by ANOVA using SAS 9.4 [29]. Following significant main effects, multiple comparisons among means were completed using the Student–Newman–Keuls test. A result of $p \leq 0.05$ was considered significant. The correlation coefficient (r) and probability (*p*) were determined by Pearson's correlation analysis using the Statistical Package in SigmaPlot 13.0 (Systat Software Inc., San Jose, CA, USA).

## 3. Results

### 3.1. Hybrid Vigour in Intra-Specific Hybrids

Preliminary analysis of the 6-year-old tree height, diameter, and stem volume data from the Al-Pac field sites, AB1 and AB2 only, indicated differences in family performance among the different cross-types. In order to detect the intra-specific hybrid vigour, intra-regional crosses (AB♀× AB♂ and QC♀× QC♂) were used as a control to compare with inter-regional (AB♀× QC♂ and QC♀× AB♂) crosses. In general, AB♀× AB♂ (the female is described first) families were the slowest growing, whereas AB × QC crosses ranked first and significantly better than intra-regional crosses in terms of growth performance (height, diameter at breast height (DBH), and stem volume) (Table 2). These results were used to make further selections for families and progeny based on cross-type for the greenhouse experiment.

**Table 2.** Mean height ($\pm$ SE), diameter at breast height (DBH), and stem volume at age six from two Alberta sites (AB1 and AB2) for the four cross-types as indicated by Alberta (AB) and Quebec (QC) with the female parent listed first. Significant differences between cross-type means are indicated by different letters.

| Cross-Types | Height (m) | DBH (mm) | Stem Volume (cm$^3$) |
|---|---|---|---|
| AB♀× AB♂ | 3.67 $\pm$ 0.05 [c] | 33.70 $\pm$ 0.76 [b] | 1375.94 $\pm$ 65.88 [b] |
| AB♀× QC♂ | 4.06 $\pm$ 0.04 [a] | 37.77 $\pm$ 0.54 [a] | 1786.78 $\pm$ 59.48 [a] |
| QC♀× AB♂ | 3.78 $\pm$ 0.03 [bc] | 34.73 $\pm$ 0.40 [b] | 1389.78 $\pm$ 35.27 [b] |
| QC♀× QC♂ | 3.80 $\pm$ 0.05 [b] | 34.52 $\pm$ 0.63 [b] | 1433.09 $\pm$ 55.63 [b] |

### 3.2. Greenhouse Growth at Two Months (34 Days after Transplanting)

Initial growth of cuttings after striking, for all families in the FG, SG, and SFG groups, showed similar heights during the first seven days of growth, early in the growing season. However, after the emergence of primary leaves, cuttings of the fast-growing group (FG) grew much faster than the

cuttings of the intermediate (SFG) or the SG group selected from the slow-growing families. By Day 34 after transplanting (about two months old), the FG group performed better than both the SG and SFG groups in stem volume (Figure 2) and biomass (Figure 3). Additionally, within the same family (i.e., family AP5416), fast-growing progeny (272084) performed better than the slow-growing progeny (272023 and 272024) under near-optimal greenhouse conditions, thus indicating the wide range of performance variability within a single family (Figure 4).

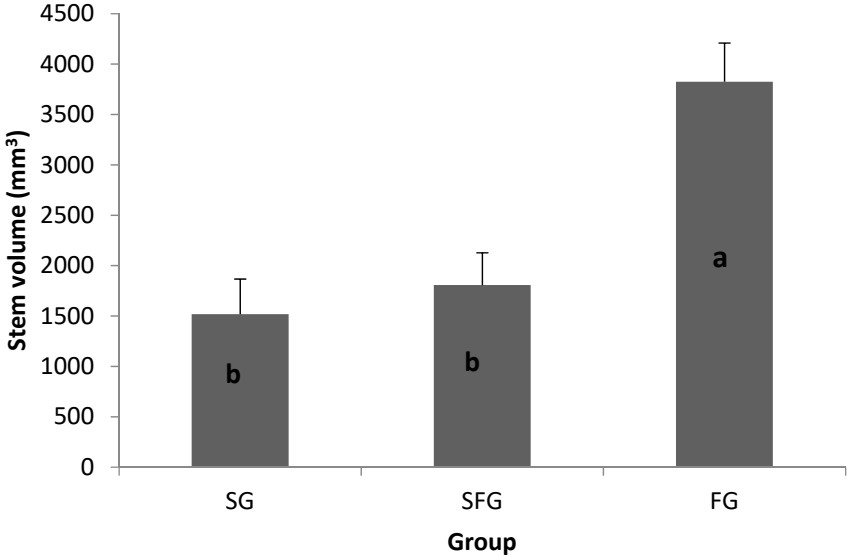

**Figure 2.** Mean stem volume (mm$^3$) (+SE) of three groups at Day 34 under near-optimal greenhouse conditions. Note: SG = slow-growing progeny; SFG = slow-growing progeny from a fast-growing family; and FG = fast-growing progeny. Significant differences between group means are indicated by different letters at $p \leq 0.05$.

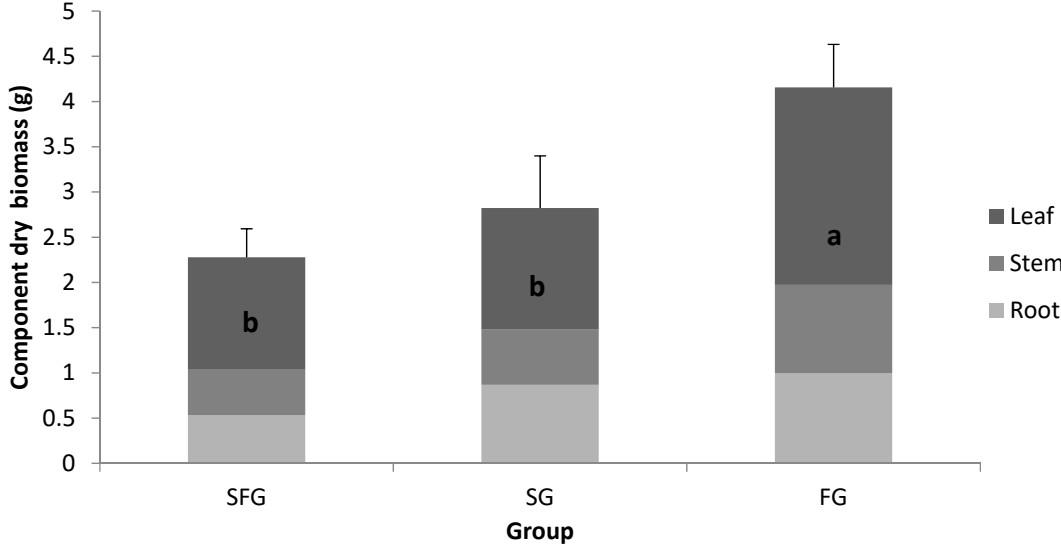

**Figure 3.** Mean component dry biomass (g) (+SE) of three groups at Day 34 under near-optimal greenhouse conditions. Note: SG = slow-growing progeny; SFG = slow-growing progeny from a fast-growing family; and FG = fast-growing progeny. Significant differences between group means of total dry mass are indicated by different letters at $p \leq 0.05$.

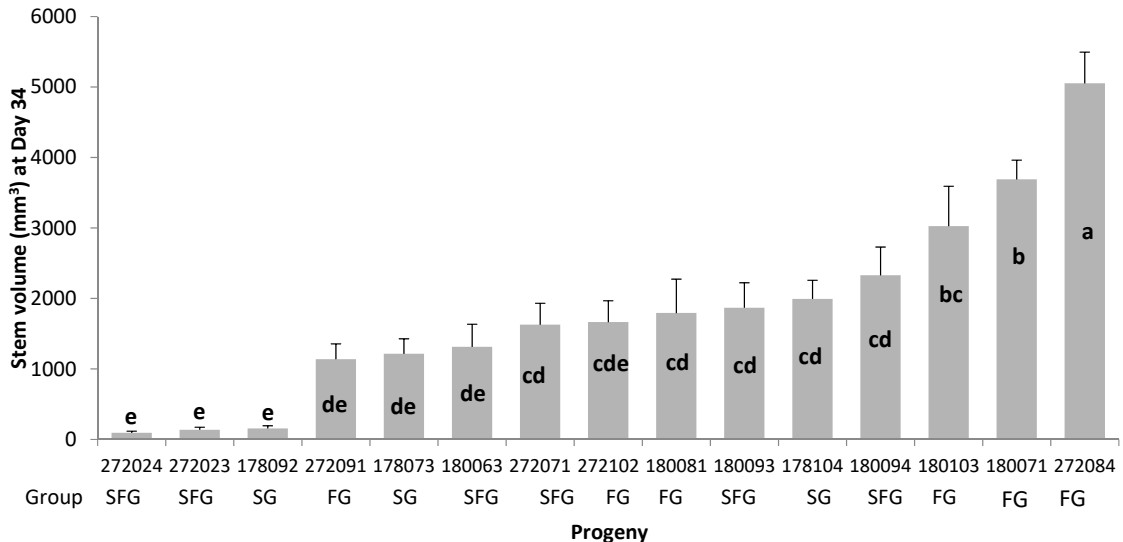

**Figure 4.** Mean stem volume (mm$^3$) (+ SE) of 15 selected balsam poplar progeny at Day 34 under near-optimal greenhouse conditions. Note: SG = slow-growing progeny; SFG = slow-growing progeny from a fast-growing family; and FG = fast-growing progeny. Significant differences between progeny means are indicated by different letters at $p \leq 0.05$.

### 3.3. Greenhouse Gas Exchange

Table 3 shows that higher rates of photosynthesis were correlated with growth (height, caliper, and stem volume) at Day 34 (after transplanting, about 2-months old), just prior to harvest. Additionally, increased photosynthetic demand was supported by an increased supply of $CO_2$ through an increase in stomatal conductance ($g_s$). It was also found that $\delta^{13}C$ in the leaf tissue showed a significant negative correlation with height which supports the $g_s$ results, indicating that the stomates were open, promoting an increase in gas exchange and carbohydrate production, resulting in an increase in height growth.

**Table 3.** Pearson's correlations (r) analysis among physiological variables under optimal growth condition in greenhouse for selected progeny growth at two months.

|  | Height (cm) | Caliper (mm) | Stem Volume (mm$^3$) | A | $g_s$ | iWUE | $\delta^{13}C_{leaf}$ (‰) |
|---|---|---|---|---|---|---|---|
| **Height (cm)** | 1 | 0.933 ** | 0.861 ** | 0.751 * | 0.532 * | 0.606 * | −0.545 ** |
| **Caliper (mm)** |  | 1 | 0.890 ** | 0.743 * | 0.614 * | 0.517 * | −0.502 |
| **Stem volume (mm$^3$)** |  |  | 1 | 0.556 * | 0.327 | 0.344 | −0.495 |
| **A** |  |  |  | 1 | 0.596 * | 0.796 ** | −0.461 |
| **$g_s$** |  |  |  |  | 1 | 0.205 | −0.563 * |
| **iWUE** |  |  |  |  |  | 1 | −0.505 * |

* Significant at $p \leq 0.05$: (1) ** Significant at $p \leq 0.01$. A: Photosynthetic rate ($\mu$mol $CO_2$ m$^{-2}$ s$^{-1}$); $g_s$: stomatal conductance (mol $H_2O$ m$^{-2}$ s$^{-1}$); iWUE: intrinsic water use efficiency ($\mu$mol $CO_2$ mmol$^{-1}$ $H_2O$); $\delta^{13}C$ leaf: carbon isotope composition.

### 3.4. Comparisons of Greenhouse Growth at Day 34 and Field Growth Performance at Age Eight Years

The Pearson's correlation analysis showed that stem volumes calculated from height and caliper of Day 34 stecklings grown under near-optimal greenhouse conditions are positively and significantly correlated with stem volumes of 6-year- and 8-year-old field-grown trees of the same genotypes (Figure 5). Additionally, positive and significant correlations were also obtained when stem dry biomass and stem volumes of Day 34 stecklings grown under near-optimal greenhouse conditions (Table 4) were regressed against each of 6- and 8-year-old stem diameters and stem volume of field-grown trees of the same genotypes.

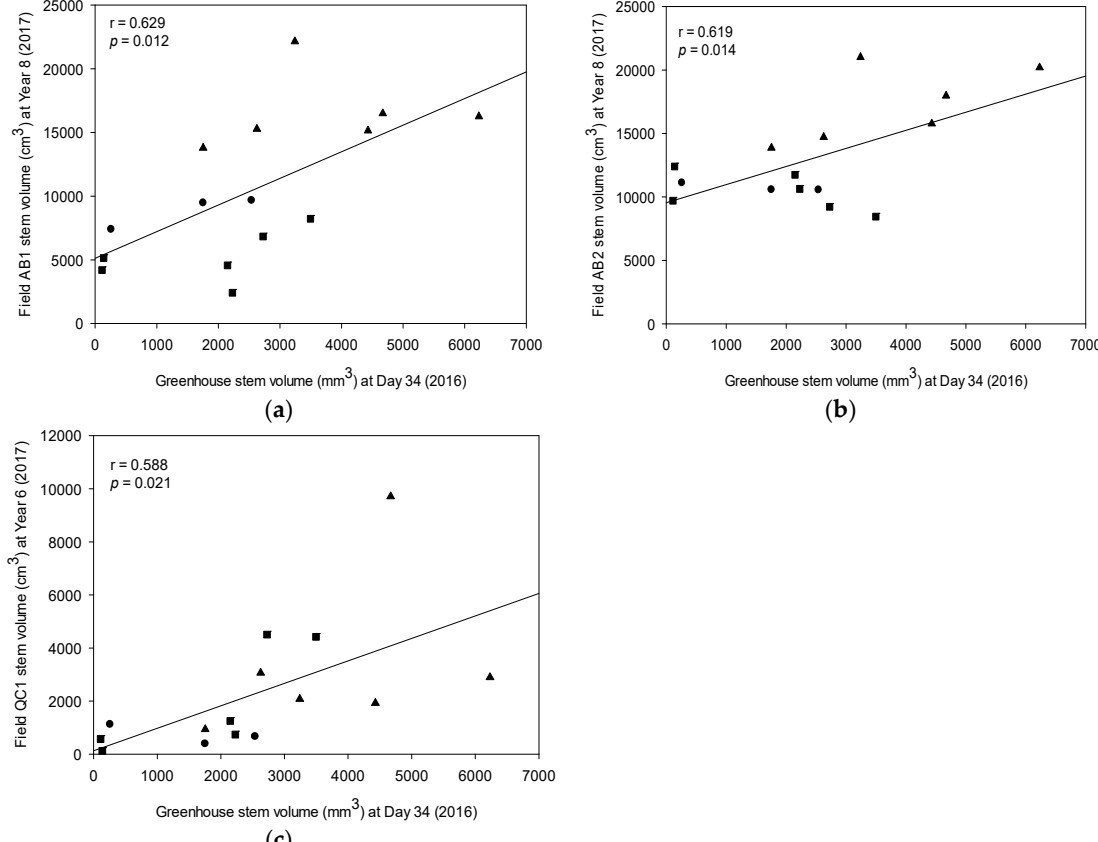

**Figure 5.** Pearson's correlation (r) between field stem volume and greenhouse growth at Day 34 (after transplanting) for 15 selected progeny in 2017 for: (**a**) Year 8, Field AB1, AB; (**b**) Year 8, Field AB2, AB; and (**c**) Year 6, Field QC1, QC. Symbols represent individual progeny that had been previously grouped with FG (▲), SFG (■), and SG (●). Note: SG = slow-growing progeny; SFG = slow-growing progeny from a fast-growing family; and FG = fast-growing progeny. AB = Alberta and QC = Quebec.

**Table 4.** Pearson's correlation (r) analysis between phenotypic characteristics of 8-year-old field-grown hybrid *Populus balsamifera* trees and Day 34 greenhouse-grown stecklings for the same 15 genotypes. AB = Alberta and QC = Quebec.

| **Field AB1** | | |
|---|---|---|
| **Stem parameters of 8-year-old trees at FieldAB1** | **Day 34 stem volume (mm³)** | **Day 34 stem dry weight (g)** |
| Height (cm) | 0.550 *[1] | 0.516 * |
| DBH (mm) | 0.608 * | 0.584 * |
| Stem volume (cm³) | 0.629 * | 0.601 * |
| **Field AB2** | | |
| **Stem parameters of 8-year-old trees at** | **Day 34 stem volume (mm³)** | **Day 34 stem dry weight (g)** |
| Height (cm) | 0.654 ** | 0.686 ** |
| DBH (mm) | 0.537 * | 0.604 * |
| Stem volume (cm³) | 0.619 * | 0.671 ** |
| **Field QC1** | | |
| **Stem parameters of 6-year-old trees at** | **Day 34 stem volume (mm³)** | **Day 34 stem dry weight (g)** |
| Height (cm) | 0.758 ** | 0.739 ** |
| DBH (mm) | 0.666 ** | 0.705 ** |
| Stem volume (cm³) | 0.588 * | 0.684 ** |

[1] All values represent the correlation coefficient (r) from Pearson's correlation analysis. DBH = diameter at breast height (1.3 m). ** Significant at $p \leq 0.01$. * Significant at $p \leq 0.05$.

### 3.5. Hormone Analysis

Since the gibberellin (GA) profiles are likely to be very different between the active growth phase and growth cessation (bud set) phase, we grew the poplar trees under near-optimal conditions and harvested the tissue near the longest day of the year to avoid the deficiency and degradation of bioactive GAs [30].

The concentrations of three endogenous plant hormone classes, namely, ABA, IAA, and GAs, were quantified in stem tissues of 15 selected FG, SFG, and SG progeny. Stem IAA concentration in Day 34 stecklings was positively and significantly correlated with stem biomass and stem volume (Figure 6a,b). In contrast, stem ABA in Day 34 stecklings was negatively and significantly correlated with stem volume and stem dry biomass (Figure 6c,d).

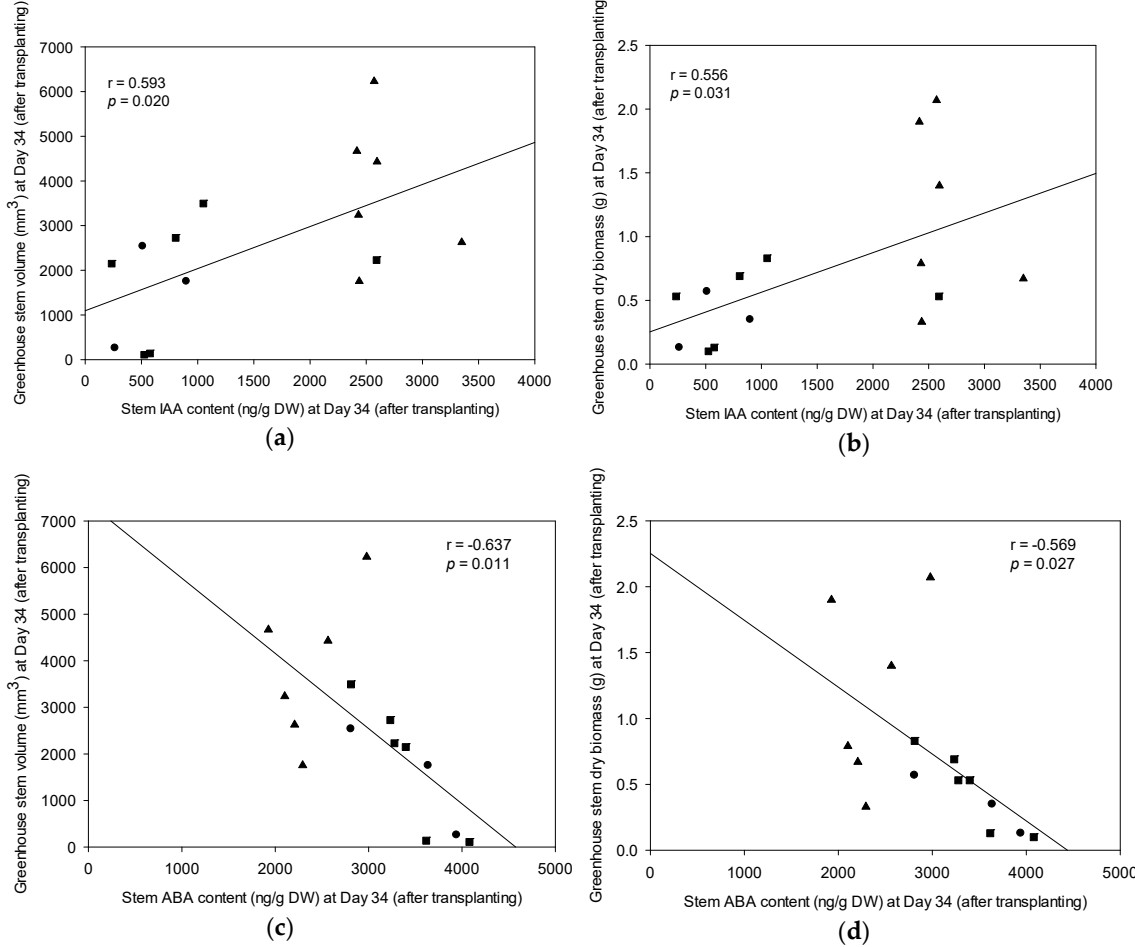

**Figure 6.** Pearson's correlations (r) of tissue concentrations of indole-3-acetic acid (IAA) and abscisic acid (ABA), versus stem dry biomass or stem volume of nursery-grown cuttings (new growth) at Day 34 (after transplanting). (**a**) New growth stem volume at Day 34 *versus* IAA in stem tissue at Day 34; (**b**) New growth stem dry biomass at Day 34 versus IAA in stem tissue at Day 34; (**c**) New growth stem volume at Day 34 *versus* ABA in stem tissue at Day 34; and (**d**) New growth stem dry biomass at Day 34 *versus* ABA in stem tissue at Day 34. Symbols represent individual progeny that had been previously grouped with FG (▲), SFG (■), and SG (●). Note: SG = slow-growing progeny; SFG = slow-growing progeny from a fast-growing family; and FG = fast-growing progeny.

### 3.6. Endogenous Plant Growth Hormone in Greenhouse-Grown Stecklings versus Field Growth Performance

The relationships between stem GA levels and stem volume were analyzed through Pearson's correlation (Figure 7). Our results showed that stem $GA_{19}$ and $GA_{20}$ content were all significant

and negatively correlated with greenhouse stem volume, which indicates that $GA_{19}$ and $GA_{20}$ serve as persecutors for $GA_8$ (Figure 7a,b). In addition, we confirmed that both $GA_{19}$ and $GA_{20}$ play an important role in the early *GA20ox* portion of the GA biosynthesis pathway as found by Ma et al. [11]. In the elongated stems of the Day 34 stecklings, a significant and positive correlation between stem volume and the content of $GA_8$ was detected, with an r of 0.840 ($p < 0.001$) (Figure 7c). Moreover, a positive and significant correlation was also detected between field growth (Fields AB1 and AB2) trees and the content of $GA_8$ in the greenhouse-grown trees for the 15 selected progeny (r = 0.749 ($p = 0.001$) and r = 0.734 ($p = 0.002$), respectively) (see Figure 8).

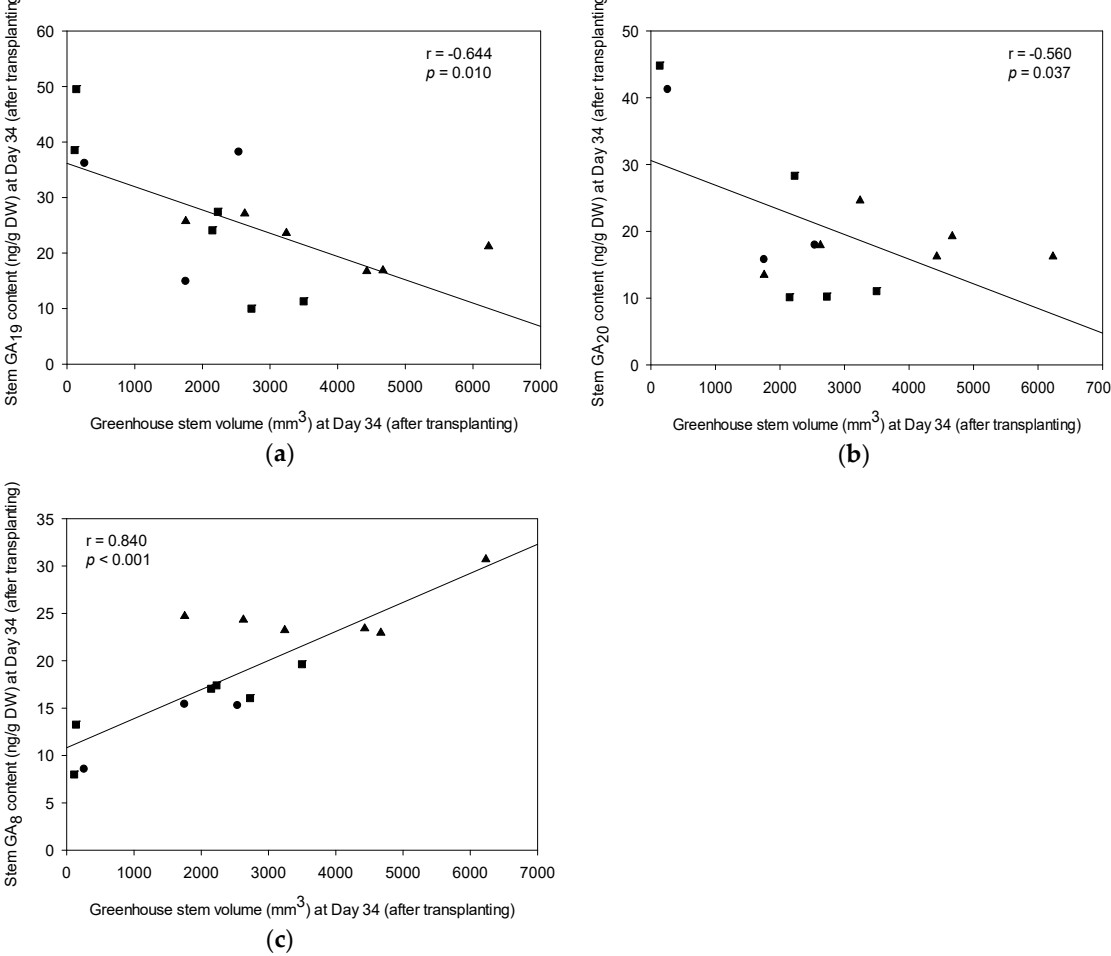

**Figure 7.** Pearson's correlation (r) between greenhouse stem volume at Day 34 (after transplanting) and plant hormones for 15 selected progeny; (**a**) $GA_{19}$ (ng/g DW); (**b**) $GA_{20}$ (ng/g DW); and (**c**) $GA_8$ (ng/g DW). Symbols represent individual progeny that had been previously grouped with FG (▲), SFG (■), and SG (●). Note: SG = slow-growing progeny; SFG = slow-growing progeny from a fast-growing family; and FG = fast-growing progeny.

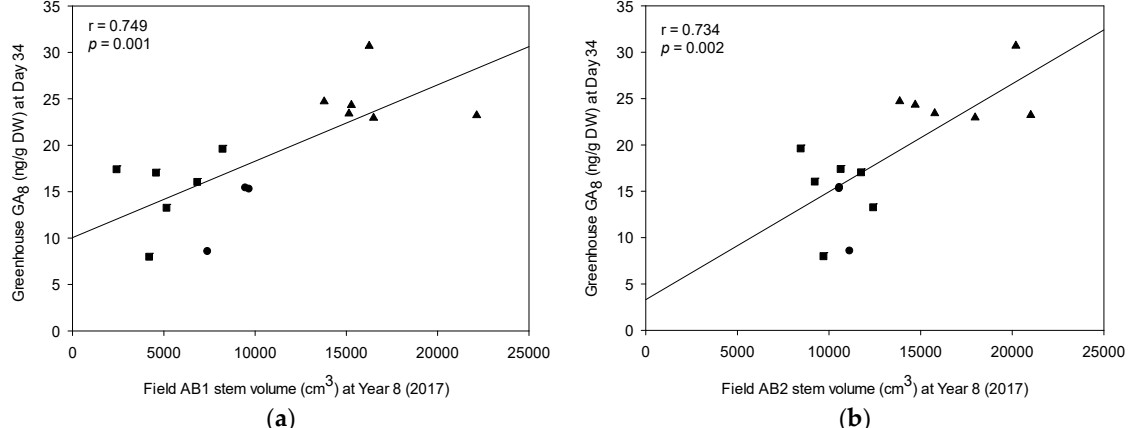

**Figure 8.** Pearson's correlation (r) between greenhouse $GA_8$ (ng/g DW) at Day 34 (after transplanting) and field stem volume for 15 selected progeny; (**a**) Field AB1, AB; and (**b**) Field AB2, AB. Symbols represent individual progeny that had been previously grouped with FG (▲), SFG (■), and SG (●). Note: SG = slow-growing progeny; SFG = slow-growing progeny from a fast-growing family; and FG = fast-growing progeny.

## 4. Discussion

Phenology is the study of the timing of yearly events such as spring bud flush, growth cessation, autumn bud set, and leaf senescence. The two major environmental cues which are primarily responsible for the induction of phenological traits are temperature and photoperiod (day length) [31]. There is an apparent trade-off between photosynthetic carbon assimilation rates (A), growth, and phenology in *P. balsamifera* [27]. If a transfer south is associated with warmer than optimal temperatures, growth cessation will be delayed and bud formation will proceed more slowly. If a transfer north is associated with colder than optimal temperatures, growth cessation will also be delayed and additionally bud formation will take longer [32]. Trees from the AB genotypes have higher A but accomplish far less growth than the trees from the QC genotypes because of the latitude differences that drive the photoperiod differences. However, if no intrinsic physiological constraints exist which might prevent the combination of high A from the AB parent and the longer growing period of the QC parent, the progeny of intra-specific crosses between AB and QC populations may accomplish more growth than local crosses (i.e., heterosis). From our results, the crossing of AB and QC *P. balsamifera* (Table 2) showed superior growth and heterosis (hybrid vigour), the phenomenon where hybrids outperform their parents in yield, biomass, and other traits [1]. Additionally, our findings showed that intra-specific hybridizations between geographically distant populations may lead to heterosis [33,34] through bringing together a new combination of alleles. However, there were no significant differences between the QC × AB cross-type compared with the local crosses, and other intra-regional crosses may indicate that the maternal effect played a role in the growth of the $F_1$ hybrids, with offspring from AB mothers exhibiting higher fitness than those from QC mothers [35,36]. This may explain why the AB × QC cross-type showed better performance than the corresponding QC × AB families.

Several past studies have indicated that morphological measures of young conifer trees grown under near-optimal conditions can be predictive of inherently rapid stem growth, at the family level, at older ages (i.e., 9 years or 32 years) [14,37]. However, our study is the first study to show a significant and positive correlation between greenhouse-grown (Day 34) and field-grown (age eight) balsam poplars, while similar findings have been found in the literature for conifers, namely, 12 open-pollinated families of *Pinus densiflora* Sieb. et Zucc., where seedling stem volume at age six months was significantly correlated with field performance at age 32 years [14]. In addition, Pharis et al. [37] found that full-sibling families *of P. radiata*, that is, seedling stem volume at age 138 days, gave reliable estimates of field performance at any age measured over nine years. Additionally, six-month heights of full-sibling families of black spruce (*P. mariana* Mill.) which were grown in a

greenhouse environment were significantly correlated with age 13 field heights [38]. From the above discussion, two common characteristics were identified. First, for all the retrospective approaches studied, the cuttings were grown under near-optimal environmental conditions in greenhouses. Second, morphological measurements were made prior to the setting of the terminal bud. Therefore, these findings imply that the early stem and shoot growth of young rooted cuttings (two months) that are raised under near-optimal conditions can be a useful trait for identifying inherently rapid stem growth in mature balsam poplar.

Leaf $\delta^{13}C$ is largely related to the ratio of $CO_2$ partial pressure inside the leaf and ambient air (ci/ca) [39], which is driven by stomatal conductance and photosynthetic processes. Several studies have shown a strong positive correlation between $\delta^{13}C$ and plant water use efficiency (WUE) via ci/ca [40–42], which suggests that leaf $\delta^{13}C$ can be measured as a proxy for plant WUE. Water use efficiency reflects the balance between carbon fixation and the amount of water released by plants. Water use efficiency trends were assessed through $\delta^{13}C$ values of leaf tissue, with higher $\delta^{13}C$ values generally associated with greater WUE, whereas more negative $\delta^{13}C$ values were linked to reduced WUE and greater water loss [43,44]. Trees displaying both high WUE and high productivity, therefore, likely indicate that WUE is primarily associated with variation in photosynthetic capacity, as opposed to variation in stomatal conductance, which would theoretically lead to a decrease in productivity. In conifer species, positive correlations between growth traits and $\delta^{13}C$ values have consistently been found [43,45], and our results, shown in Table 3, also indicate the same trend in a deciduous species, suggesting variation in WUE is driven primarily by photosynthetic capacity. A high sink demand resulting from increased growth leads to an increase in photosynthetic rate, and, subsequently, more positive $\delta^{13}C$ values and higher WUE.

High IAA levels have been associated with enhanced levels of expression of two GA biosynthesis genes, *GA20ox* and *GA3ox* [46–48]. Thus, biosynthesis of the GAs required for rapid stem growth may be dependent on an adequate supply of the auxin IAA. In our study, stem tissue ABA concentrations in Day 34 stecklings showed a significant negative correlation with stem volume and dry biomass (Figure 6c,d); thus, an ABA:IAA "balance" may control very early height growth in rooted cuttings. Recent evidence suggests that IAA acts downstream of the ABA response [49], and there is also the potential for high ABA concentrations to negatively influence the biosynthesis of GAs [50], including inhibition at the gene expression level.

Elevated levels of bioactive GA usually suppress the expression of *GA20ox* and *GA3ox* while stimulating the expression of *GA2ox*; conversely, a drop in the bioactive GA level usually up-regulates the expression of *GA20ox* and *GA3ox* and down-regulates the expression of *GA2ox* [11,51]. As the concentration of growth-activating GAs in growing tissues is controlled by the transcriptional control of biosynthetic (*GA20ox* and *GA3ox*) or catabolic (*GA2ox*) genes [13], the modification of the expression of these genes alters plant growth, which has been successfully demonstrated in various plant species including poplars [52–54]. The trend shown in Figure 8 indicates the possibility of using endogenous GA levels to accelerate the early selection of balsam poplars that possess traits for inherently rapid stem growth. Moreover, the correlations found between stem volume and GA content in our study confirm the findings of others such as Zhang et al. [55] who found that GA content was correlated with heterosis in plant height in wheat hybrids where it was reported that increased elongation of the uppermost internode contributed most to heterosis for plant height. In the case of *Populus*, fiber and vessel lengths increase in the transition zone between juvenile and mature wood [56]. Therefore, the time it takes for a cell to mature within the different differentiation zones will ultimately determine its size and cell wall thickness. It is possible that the action of GAs like $GA_1$ and/or $GA_4$ extends this transition time, and thereby increases the fiber length in hybrid aspen (*Populus tremula* $\times$ *P. tremuloides*) compared to pure *P. tremula* plants [18]. Additionally, these results also provide new insights into the mechanisms whereby GAs control growth and development in trees. Overexpression of *Arabidopsis* GA 20-oxidase (*AtGA20ox1*) in transgenic hybrid aspen (*Populus tremula* $\times$ *P. tremuloides*) resulted in a substantial increase in plant height and fiber length, nearly doubling stem dry weight relative to

the wild-type control. Additionally, *GA20ox* appears to play an important role in wood development through tree growth. Ko et al. [57] showed that poplar GA20-oxidase (*PtGA20ox*) has the highest expression in developing xylem, where several GA receptors were highly expressed in the tissue by using tissue-specific transcriptome analysis.

## 5. Conclusions

In conclusion, the data presented in this research confirms the widespread occurrence of heterosis typically found in interspecific crosses, with disparate population breeding of intra-specific hybrids in balsam poplar. The combined analyses of endogenous GA content and growth data revealed that GAs play a regulatory role in heterosis for hybrid balsam poplar at the physiological level. Larger scale investigations of multiple plant hormones at more developmental stages of hybrid balsam poplar are anticipated to confirm and extend these findings. Moreover, the results obtained from selected progeny are validated by retrospective comparisons with the stem growth performance of the corresponding field-grown trees at eight years of age. These results with *P. balsamifera* also point toward the successful early selection of progeny from genetic crosses of other deciduous tree species, that is, the identification of progeny which are inherently fast growing and have high stem biomass production. Additionally, the relationship between GAs and growth performance (both field and greenhouse) may open up new ways to manipulate trees to grow faster and produce more biomass by increasing endogenous GA levels. Moreover, screening for high levels of expression of GA20ox genes may be a useful technique for future tree-breeding programs. Future RT-PCR work should be used to test *GA20oxs*, *GA3oxs*, and *GA2oxs* which are the main targets of regulation by GA signaling to establish homeostasis.

**Author Contributions:** Y.H. and B.R.T. conceived and designed the experiments; performed the experiments; analyzed the data; and wrote the paper.

**Funding:** This research was funded by a Natural Sciences and Engineering Research Council of Canada (NSERC) Industrial Research Chair in Tree Improvement to B.R. Thomas, grant number 474636-13.

**Acknowledgments:** We are grateful to Pierre Périnet (QC) who conducted the breeding, David Kamelchuk (Al-Pac), Michael Thomson, Briana Ledic, Gregg Hamilton, and Ho-Chun (Aaron) Chen for their assistance in collecting and processing plant material, establishing field trials, and offering assistance with data collection. We also thank Steven Williams (retired), University of Alberta, and Loeke Janzen for the hormone analysis, University of Calgary. Funding for this manuscript was provided through the Industrial Research Chair in Tree Improvement held by B.R. Thomas supported by the Natural Sciences and Engineering Research Council (NSERC), Alberta-Pacific Forest Industries Inc., Alberta Newsprint Company Timber Ltd., Canadian Forest Products Ltd., Millar Western Forest Products Ltd., Huallen Seed Orchard Company Ltd., West Fraser Mills Ltd. (including Alberta Plywood, Blue Ridge Lumber Inc., Hinton Wood Products, and Sundre Forest Products Inc.), and Weyerhaeuser Company Ltd., (Pembina Timberlands and Grande Prairie Timberlands). Additional funding was also provided by the Government of Alberta. Finally, we want to offer our special thanks and dedicate this paper to (Dick) R.P. Pharis who, although no longer with us, continues to inspire us by his knowledge and love of research.

**Conflicts of Interest:** The authors declare no conflict of interest.

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
