# Peer review of "Hormones and Heterosis in Hybrid Balsam Poplar (Populus balsamifera L.)"

_forests, doi:10.3390/f10020143_

Round 1

Reviewer 1 Report

Corrections needed to be made in the manuscript:

1. The description of gibberellic acid metabolism genes in the Introduction is too extensive and some of the material should be referenced in the Discussion. The role of IAA and ABA in plant growth and development in the Introduction is very poor.

2. In Figures 2-4 there are no significant differences between group and progeny means.

3. The section on GA gene expression is better to move from the Conclusion to the Discussion.

Author Response

Response to Reviewer 1 Comments:

Thank you for your review of our paper. We have answered each of your points below.

Point 1: The description of gibberellic acid metabolism genes in the Introduction is too extensive and some of the material should be referenced in the Discussion. The role of IAA and ABA in plant growth and development in the Introduction is very poor.

Response 1: Moved some of the description of gibberellic acid metabolism genes’ information into the discussion, and added information for the role of IAA and ABA in plant growth and development.

Point 2: In Figures 2-4 there are no significant differences between group and progeny means.

Response 2: Results of statistical analysis were added into the figures. Significant differences between group and progeny means are indicated by different letters.

Point 3: The section on GA gene expression is better to move from the Conclusion to the Discussion.

Response 3: Moved the section on GA gene expression from Conclusion to the Discussion.

Reviewer 2 Report

Need to reduce the conclusion in one paragraph 

Name place typing and formatting mistakes need to correct

Author Response

Response to Reviewer 2 Comments:

Thank you for your review of our paper. We have answered each of your points below.

Point 1: Need to reduce the conclusion in one paragraph

Response 1: Reduced the conclusion in one paragraph

Point 2: Name place typing and formatting mistakes need to correct

Response 2: Corrected the name place typing and formatting mistakes

Table 1. missing information.

Information is not missing here as for slow family, we only select 3 active growing progeny to be our slow growing group, therefore, no progeny for fast growing group.

Line 438-441, missing information.

Section is updated.
